# Influence of acidity on liquid−liquid phase transitions of mixed SOA proxy−inorganic aerosol droplets

Yueling Chen[1] & Xiangyu Pei[1], Huichao Liu[1], Yikan Meng[1], Zhengning Xu[1], Fei Zhang[1], Chun Xiong[1], Thomas C. Preston[3], Zhibin Wang[1,2,4*]

[1]College of Environmental and Resource Sciences, Zhejiang Provincial Key Laboratory of Organic Pollution Process and Control, Zhejiang University, Hangzhou 310058, China
[2]ZJU-Hangzhou Global Scientific and Technological Innovation Center, Zhejiang University, Hangzhou 311215, China
[3]Department of Atmospheric and Oceanic Sciences and Department of Chemistry, McGill University, 805 Sherbrooke Street West, Montreal, Quebec H3A 0B9, Canada
[4]Key Laboratory of Environment Remediation and Ecological Health, Ministry of Education, Zhejiang University, Hangzhou 310058, China

*Correspondence to*: Zhibin Wang (wangzhibin@zju.edu.cn)

Yueling Chen and Xiangyu Pei contribute equally to this work.

**Abstract.** Phase state and morphology of aerosol particles play a critical role in determining their effect on climate. While aerosol acidity has been identified as a key factor affecting the multiphase chemistry and phase transitions, the impact of acidity on phase transition of multicomponent aerosol particles has not been extensively studied in situ. In this work, we employed an aerosol optical tweezer (AOT) to probe the impact of acidity on the phase transition behavior of levitated aerosol particles. Our results revealed that higher acidity decreases the separation relative humidity (SRH) of aerosol droplets mixed with ammonium sulfate (AS) and secondary organic aerosol (SOA) proxy, such as 3-methylglutaric acid (3-MGA), 1,2,6-hexanetriol (HEXT) and 2,5-hexanediol (HEXD) across aerosol pH in atmospheric condition. Phase separation of organic acids was more sensitive to acidity compared to organic alcohols. We found the mixing relative humidity (MRH) was consistently higher than the SRH in several systems. Phase-separating systems, including 3-MGA/AS, HEXT/AS, and HEXD/AS, exhibited oxygen-to-carbon ratios (O:C) of 0.67, 0.50, and 0.33, respectively. In contrast, liquid-liquid phase separation (LLPS) did not occur in the high O:C system of glycerol/AS, which had an O:C of 1.00. Additionally, the morphology of 42 out of the 46 aerosol particles that underwent LLPS was observed to be a core-shell. Our findings provide a comprehensive understanding of the pH-dependent LLPS in individual suspended aerosol droplets and pave the way for future research on phase separation of atmospheric aerosol particles.

## 1 Introduction

Atmospheric aerosol particles can directly and indirectly impact climate by absorbing and scattering light and acting as cloud condensation nuclei (Rosenfeld et al., 2014). Particle morphology is a critical factor influencing the physiochemical properties

of aerosols such as their optical properties, chemistry, and nucleation processes (Freedman et al., 2009; Corral Arroyo et al., 2022; Cosman et al., 2008; Lam et al., 2021; Petters and Kreidenweis, 2007; Mikhailov et al., 2021). Morphology can be broadly categorized into single-phase homogeneous morphology and phase separation morphology (Bertram et al. 2011; Ciobanu et al. 2009), based on the phase state of the particle. For droplets with a phase separation morphology, the two main equilibrium morphologies are a fully engulfed (core-shell) structure and a partially engulfed structure (Reid et al. 2011). Droplets can undergo phase transition processes and thus the morphology would be changed. The composition and mass of inorganic and organic components impact the phase transition characteristics of a particle. With a decrease of particle water content, a transition occurs from single homogenous liquid phase to two separated liquid phases, which is known as liquid-liquid phase separation (LLPS; Freedman et al., 2017). The relative humidity (RH) when the LLPS occurs is defined as separation relative humidity (SRH). The reverse process, in which two liquid phases mix into a single homogenous liquid phase, is referred to as liquid-liquid phase mixing and the corresponding RH is the mixing RH (MRH; You et al., 2014; Gorkowski et al., 2017).

The phenomenon of LLPS has garnered considerable attention from the atmospheric research community due to its potential role in affecting the physiochemical properties of atmospheric aerosols (Kucinski et al., 2019; Ott et al., 2020; Freedman, 2020). Song et al. (2012) using optical microscopy studied the relationship between LLPS and the oxygen-to-carbon ratio (O:C) and discovered that LLPS was consistently observed when O:C < 0.56, while it was never observed when O:C > 0.80. For O:C between 0.56 and 0.80, the occurrence of LLPS was influenced by the types of organic functional groups. Gorkowski et al. (2020) utilized experimental results of previous studies on LLPS and morphology, observing a general trend in morphology from partially engulfed to core shell and finally homogeneous as oxidation increased. More recently, it is found that submicrometer-sized aerosol particles had a lower SRH compared to micrometer-sized droplets (Kucinski et al., 2021; Ohno et al., 2021). Meanwhile, Stewart et al. (2015) employed aerosol optical tweezer (AOT) to investigate the morphologies of aqueous droplets. They found in the polyethylene glycol (PEG)/ammonium sulfate (AS) system, droplets formed predominately core-shell particles when the AS content was high and partially engulfed when the PEG content was high.

One factor that could influence the phase transitions of aerosol particles is the aerosol pH. The pH values for misty cloud and fog droplets generally range between 2 and 7, whereas continental and marine aerosol particles exhibit a wider range of pH values, from -1 to 5 and 0 to 8, respectively (Pye et al., 2020; Angle et al., 2021; Weber et al., 2016; Tilgner et al., 2021; Zheng et al., 2020). Meanwhile, aerosol pH is size-dependent, with the fine mode showing lower 1–4 pH units than the coarse mode (Fang et al., 2017; Young et al., 2013; Guo et al., 2017). Losey et al. (2018) studied six organic components and discovered that phase separation may be hindered by the addition of sulfuric acid, while the SRH of 3-methylglutaric acid/ammonium sulfate system was found to decrease with the addition of sodium hydroxide (Losey et al. 2016), as the deprotonation of organic component or difference in salting out ability of inorganic may change the SRH. More recently, Tong et al. (2022) investigated

the effect of acidity on phase separation in single suspended microdroplets using AOT. Their results showed that the pH can affect the miscibility of the mixture and high acidity results in a reduced SRH of 1,2,6-hexanetriol.

Our aim with this work is to gain a comprehensive understanding of the influence of pH on phase transitions in suspended droplets. To that end, we investigated pH-dependent SRH and MRH, as well as morphologies of aqueous droplets using AOT, meanwhile discussed the effect of O:C on phase separation behavior. Compared to substrate-based measurement techniques, AOT can suspend droplets without any substrate contact, providing a more realistic simulation of the behavior of aerosols in the atmosphere (Wang et al., 2021; Cui et al., 2021; Redding et al., 2015; Gong et al., 2018; Rafferty et al., 2023). We measured droplets containing AS and a range of organic compounds with varying O:C. Our findings provide insight into the mechanisms behind pH-dependent phase transitions in levitated droplets, and have implications for fields such as climate science. Overall, our study highlights the importance of considering pH as a key factor in the phase transition behavior of micron-sized droplets and underscores the need for further research to fully understand the complex interactions between pH and phase transitions in these atmospherically relevant systems.

## 2 Methods

### 2.1 Aerosol generation

Four organics components: glycerol (GL), 3-methylglutaric acid (3-MGA), 1,2,6-hexanetriol (HEXT), and 2,5-hexanediol (HEXD), were chosen as they are commonly-used secondary organic aerosol (SOA) proxies (Lam et al., 2021; Gorkowski et al., 2020). O:C of the selected chemicals varied from 1 to 0.33 (**Table 1**), which is similar to the real atmospheric SOA (Canagaratna et al., 2015; Mahrt et al., 2021). AS was chosen as the inorganic salt component due to its widespread occurrence in the atmospheric environment. All concentrations of organics and AS in the mother solutions were 50 g/L. The pure organic and inorganic components were dissolved in ultrapure water (Millipore, resistivity of 18.2 MΩ) to create solutions with organic-to-inorganic mass ratio (OIR) of 1:1. The pH of studied systems were adjusted within the range of 0.48 to 6.53 by using either concentrated sulfuric acid (SA) or sodium hydroxide (NaOH) solution (5.29 mol/L). Sodium hydroxide, a strong base, allowed for pH adjustment with minimal usage (Losey et al., 2016). However, it is necessary to acknowledge that the addition of NaOH changed the composition of the inorganic part of the solution, potentially affecting the SRH values measured. The pH of each solution was measured using a pH meter (Mettler Toledo Instruments Co., Ltd., Shanghai, China). The purity and supplier of the compounds used in this study are summarized in **Table S1**.

**Table 1.** Information of the solutions used to generate aerosol droplets.

| Solution ID | Organic component | O:C ratio | pH |
|---|---|---|---|
| GL | glycerol | 1.00 | 5.24±0.01 |
| | | | |
| 3-MGA-I | | | 0.48±0.01 |
| 3-MGA-II | | | 1.19±0.01 |
| 3-MGA-III | 3-methylglutaric acid | 0.67 | 2.70±0.01 |
| 3-MGA-IV | | | 3.70±0.01 |
| 3-MGA-V | | | 5.21±0.02 |
| 3-MGA-VI | | | 6.53±0.02 |
| | | | |
| HEXT-I | | | 0.92±0.01 |
| HEXT-II | 1,2,6-hexanetriol | 0.50 | 2.02±0.01 |
| HEXT-III | | | 3.14±0.01 |
| HEXT-IV | | | 5.11±0.02 |
| | | | |
| HEXD-I | | | 1.39±0.01 |
| HEXD-II | | | 2.03±0.01 |
| HEXD-III | 2,5-hexanediol | 0.33 | 2.71±0.01 |
| HEXD-IV | | | 3.13±0.01 |
| HEXD-V | | | 5.01±0.01 |

 **2.2 Experimental setup**

A schematic illustration of the experimental setup is presented in **Fig. S1**. The aerosol optical tweezer system consists of a custom-made levitation chamber that integrates the optical trapping system, the illumination and imaging system, and the aerosol generation system. A 532 nm (Opus 532-2W) laser was used to create an optical trap with a 100x oil immersion objective (Olympus, UPLFLN100XO, NA 1.30) pressed against a glass coverslip (Nest, thickness 160-190 μm). The illumination and imaging system includes a 450 nm LED (Daheng Optics, GCI060404) and a camera (Thorlabs, CS165CU/M) to illuminate and image the particle. Two low pass filters (Andover, 500FL07-25) were used in front of the camera lens to remove the influence of back scattered light of the 532 nm laser to photograph clear image of the particle. The Raman scattered light passed through two 50:50 beam splitters (CVI Laser Optics, BTF-VIS-50-2501M-C) and a notch filter (Semrock, NFD01-532-25x36) and was focused into the Raman spectrograph. A spectrograph (ZOLIX, Omni-λ5004i) was used to measure the Stokes shifted Raman spectrum. A 20 μm entrance slit width and 1200 groove/mm diffraction grating with a blaze wavelength of 500 nm were used to achieve a spectral resolution of 0.021 nm. The wavelength position of spectrograph was calibrated with an Hg-laser. The Raman scattered light was recorded every 4 seconds with range of 624.24-665.40 nm.

As droplets are introduced continuously into the chamber from a medical nebulizer (LANDWIND, PN100), smaller droplets
undergo a process of collision and coalescence, leading to the formation of larger droplets that can be readily trapped near the
focal point of the laser. In most cases, droplets can be successfully captured within 30 s after the introduction of an aerosol
plume into the cell. Air with relative humidity (RH) of 100% and 0% were mixed to produce wet air with a desired RH. The
flow rates of the humidified and dry air streams were regulated by mass flow controllers (MFCs, Tianjin Gastool Instruments
Co., Ltd., Tianjin, China, GT130D), with a combined flow rate of 0.3 L/min in total. Two humidity sensors (Sensirion, SHT85)
were utilized, with a precision of ±1.5%. Since the sensor located behind the chamber was positioned in close proximity (~80
mm) to the droplet, its observed values were used as a surrogate for measuring the RH inside the chamber. The RH values
were reduced in increments of 5% every 30 minutes (Tong et al., 2022; Stewart et al., 2015) until droplet phase separation
occurred. The measured values of RH given by the sensors were used as the phase separation RH. Subsequently, the RH level
was set to 100%, to investigate the phase mixing of the droplets. The entire experiment was repeated 2-4 times for each system.
**2.3 Determination of phase transitions**
When a transparent or weakly absorbing spherical particle is trapped, it can behave as a high-quality factor optical cavity that
supports sharp optical resonances, resulting in cavity-enhanced Raman scattering. These resonances can be observed as peaks
in the Raman spectrum of a particle and are often referred to as whispering gallery modes (WGMs). In principle, particle
morphology can be deduced from the WGMs, as inhomogeneities in the refractive index can disrupt the circulation of the
WGMs (Lin et al., 1992; Mitchem et al., 2006). Raman spectra measurements of single droplets in various morphological
states are presented in **Figure 1**. When the droplet was in a homogeneous phase morphology, the droplet acted as a high-
quality microcavity and sharp WGM peaks overlapped with the spontaneous Raman spectrum (**Fig. 1a**). When the droplet was
in a state of a core-shell structure, observed WGMs were clearly diminished in measured spectra (**Fig. 1b**). The origin of the
damping of the WGMs is the radial homogeneity that is present when the particle is separated into a hydrophilic core and a
hydrophobic shell. As a result, when fitting the Raman spectra with the Mie scattering model for homogeneous droplets, the
error in the best-fits greatly increase. Examination of the retrieved radius and refractive index reveals a clear break with fits
for that of a homogeneous sphere. Therefore, the point at which a significant break in particle size and refractive index occurred
can be used as the point at which core-shell phase separation occurs. As illustrated in **Fig. 1c**, when the droplet was partially-
engulfed and non-spherical, WGM peaks in the spectrum were absent (Reid et al., 2011). The origin of the spontaneous Raman
peaks at 3300 cm$^{-1}$ and ~3050 cm$^{-1}$ are identified as the spurious or weakened WGM peaks and the vibration of N-H bond,
respectively. Overall, the results of this analysis demonstrate the dynamic changes in the Raman spectra of single droplets as
they undergo morphological transitions (Sullivan et al., 2020; Stewart et al., 2015; Tong et al., 2022).

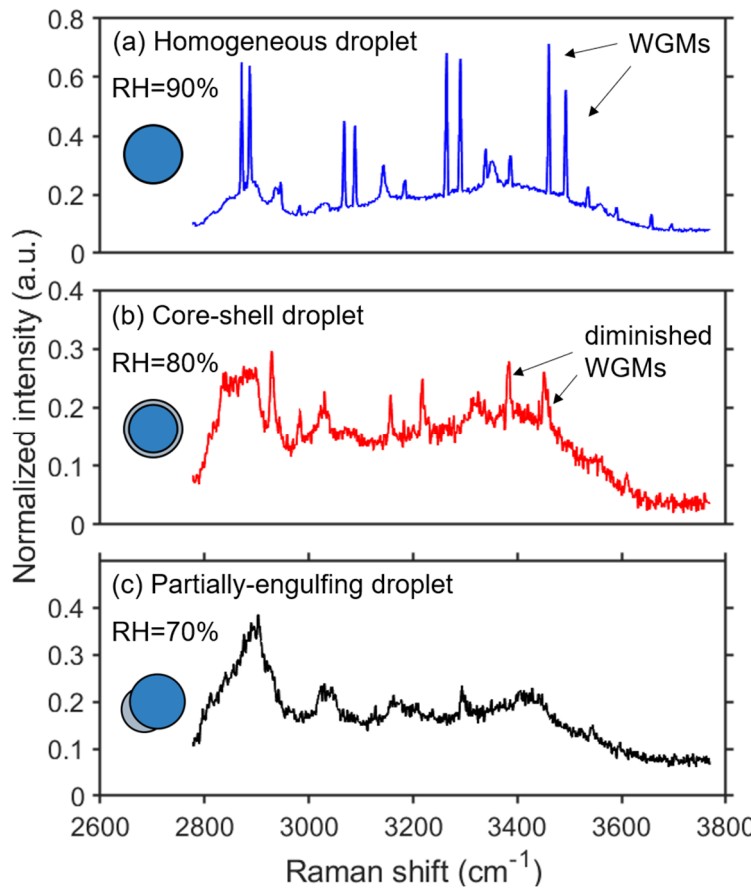

**Figure 1.** Raman spectra of 3-MGA-II microdroplets: (a) a homogenous droplet (RH = 90%); (b) a core-shell droplet (RH = 80%); (c) a partially-engulfed droplet (RH=70%). The WGMs are marked by black arrows. The normalization of the peak is achieved by dividing it by the maximum value of the spectrum's intensity, respectively.

The peak finding method used in this study is based on the ipeak code developed by O'Haver (2022). In short, the code first smooths the first derivative of the signal and identified downward-going zero-crossings that met a certain predetermined minimum slope and amplitude threshold. By adjusting the corresponding parameters, it is possible to accurately detect the desired peaks. The algorithm used to fit WGM peaks in spectra from homogenous spheres in this study was proposed by Preston and Reid (2013) and Preston and Reid (2015). The algorithm compares observed peak positions to expected positions calculated using a resonance condition from Mie theory. Error is minimized by varying particle size and refractive index (i.e. the parameters of best-fit). The method has been demonstrated to provide a rapid determination of the fitted radius and refractive index with an accuracy of ±2 nm and ±0.0005, respectively. During the experiment with reduced RH, we had to adjust the laser power to ensure the stable capture of droplets, which will affect the peak intensity. To eliminate this effect, as

demonstrated by Tong et al. (2022), we normalized all Raman spectra used in this study by the area below the spontaneous
Raman signals.

## 3 Results and discussion

### 3.1 Phase behaviors of droplets mixed SOA proxy with AS

**Fig. 2** presents the results of time-resolved Raman spectra of aerosol droplets produced from a 3-MGA-II solution under
continuously varying RH, as well as the corresponding particle size and refractive index values. To enable temperature and
RH to stabilize, the chamber was conditioned with airflow for 50 minutes after trapping a particle. During the dehumidification
process, the particle diameter decreased from 11.85 μm to 9.03 μm and the refractive index increased from 1.379 to 1.475
when RH decreased from 93.0% to 70.0%. The particle size and water content decreased with RH due to the equilibrium
partitioning of water molecules between vapor and droplets. Meanwhile, the refractive index of the droplets gradually increased
as the water content decreases. When LLPS occurred, the droplets changed from a symmetrical homogeneous phase to either
an asymmetrical partially engulfed structure which led to the disappearance of the WGMs, or the formation of a core-shell
structure. As RH in the reaction chamber was reduced, the LLPS was initiated, marked by the variations of the WGM signal
(See **Fig. 1b**). This was achieved by reducing setting RH (setting values) by 5% at 30-minute intervals until the organic phase
separated from the water-rich phase and then continuing decreasing RH by 10%-15%. **Fig. 2a** illustrates how the fitting of the
droplet diameter and the refractive index deteriorated as the shell develops, indicating phase separation. The refractive index's
shift results from a significant change in the radial profile due to the formation of a core-shell structure. Additionally, the
persistence of strong WGMs indicates that the morphology of the droplet remains spherical following LLPS and is core−shell.
During the RH increased from 70% to 95%, the reappearance of the continuously shifting WGM signal was observed,
suggesting that the inorganic phase has mixed with the organic phase, and droplet returned to a homogeneous phase.   During
the humidification process, there is an opposite trend observed in the particle size and refractive index of the droplet compared
to the dehumidification process.   In conclusion, the variations of the WGM signal can serve as a reliable indicator of the
occurrence of liquid-liquid phase separation or mixing, and the RH at these points can be considered as the SRH or MRH,
respectively. The observed phase transitions of droplets produced from HEXT-IV and HEXD-V solutions are shown in **Fig. 3**
and **Fig. 4** respectively.

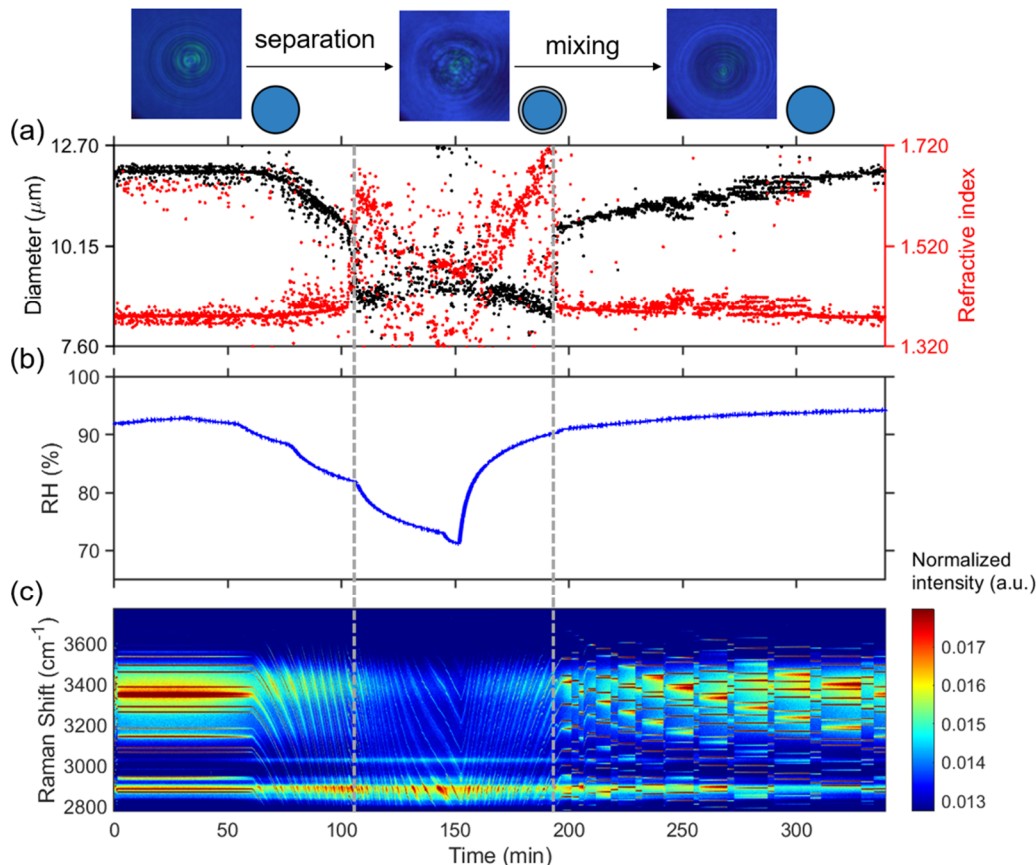

Figure 2. Liquid-liquid phase separation and mixing of aqueous 3-MGA-II. Schematic diagram of phase states is on the top of the figure. (a) Timescale of changes in droplet size and refractive index, determined from fitting the Raman shift positions of the WGMs. (b) RH variation after the trapping chamber during the humidity changing process. (c) Time-resolved Raman spectra. The cessation of the random motion of inclusions within the droplet and the resultant formation of a core-shell structure are indicated by the grey dashed line on the left. The grey dashed line on the right serves as an indication of the point at which the droplet morphology transited from a state of separated phases to a homogeneous phase. The Raman spectra at 53 min, 113 min, 130 min are shown in **Fig. 1(a), (b), (c)**, respectively. Fitting errors of the WGMs was presented in **Fig. S3**.

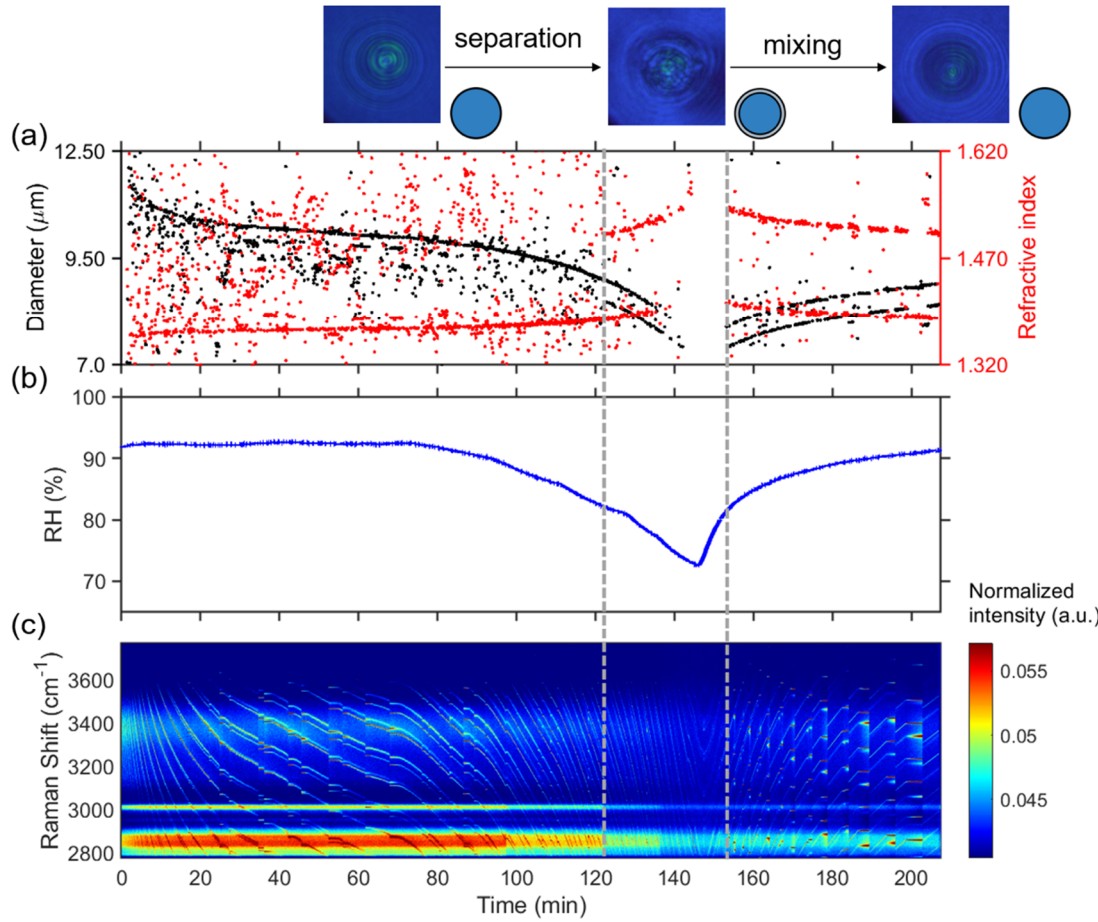

186

**Figure 3.** Liquid-liquid phase separation of aqueous of HEXT-IV. (a) Timescale of changes in droplet size and refractive index, determined from fitting the Raman shift positions of the WGMs. (b) RH variation after the trapping chamber during the humidity changing process with time. (c) Time-resolved Raman spectra. The cessation of the random motion of inclusions within the droplet and the resultant formation of a core-shell structure are indicated by the grey dashed line on the left. The grey dashed line on the right serves as an indication of the point at which the droplet morphology transitions from a state of phase separation to a homogeneous phase morphology. This transformation is characterized by the occurrence of phase mixing.

193

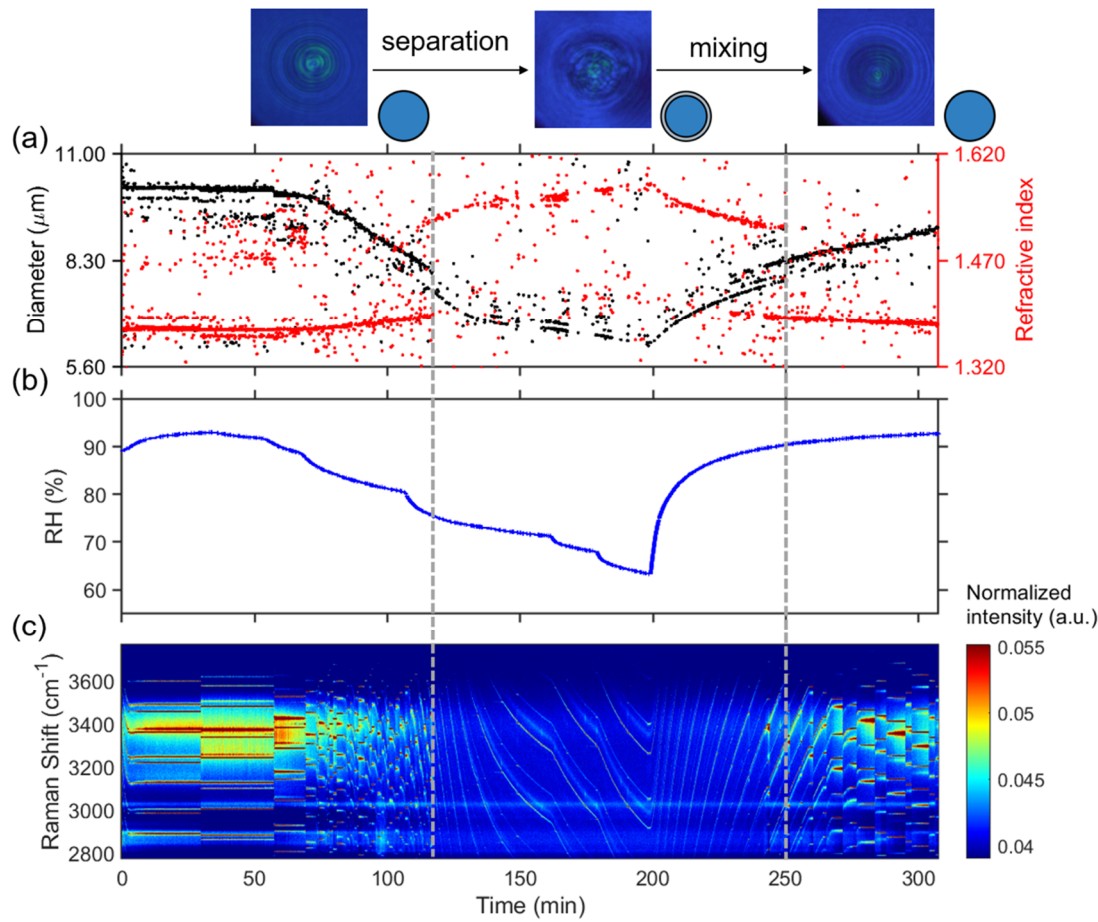

194

**Figure 4.** Liquid-liquid phase separation of aqueous of HEXD-V. (a) Timescale of changes in droplet size and refractive index, determined from fitting the Raman shift positions of the WGMs. (b) RH variation after the trapping chamber during the humidity changing process. (c) Time-resolved Raman spectra. The cessation of the random motion of inclusions within the droplet and the resultant formation of a core-shell structure are indicated by the grey dashed line on the left. The grey dashed line on the right serves as an indication of the point at which the droplet morphology transitions from a state of phase separation to a homogeneous phase morphology. This transformation is characterized by the occurrence of phase mixing.

**Fig. S2** presents the results of time-resolved Raman spectra of aerosol droplets produced from GL/AS solution under continuously varying RH, as well as the corresponding particle diameter and refractive index values. At the start of the experiment, the chamber RH was held at 93% for approximately 75 minutes. The spectrum during this period revealed a clear bright trend, indicative of the presence of many WGMs in the newly captured droplets. As the chamber RH dropped to a minimum value of 71.5% at around 200 minutes, the position of the WGMs in each spectral snapshot shifted continuously, following the same trend as the chamber RH. This observation suggests that the droplet was homogeneous and that no phase

separation occurred in the experimental RH range. The phenomenon regarding the GL/AS system is consistent with the
conclusion by Song et al. (2013) and Gorkowski et al. (2020).

## 3.2 Effect of pH on SRH and MRH of different systems

The SRH and MRH of aerosol droplets produced from 3-MGA-I~VI solution are shown in **Fig. 5a** and **Table S2**. The pH of
the 3-MGA/AS solution without the addition of an acid or base was 2.70. For solutions with a lower pH (1.19 and 0.48), SA
was added, while NaOH was added to solutions with a higher pH (3.70, 5.21, and 6.53) to adjust their pH levels. The SRH
values were 92.7%, 89.5%, 80.6%, 79.7%, 76.2% and 69.7% at pH of 6.53, 5.21, 3.70, 2.70, 1.19 and 0.48, respectively. It is
worth mentioning that when the pH of the 3-MGA system is 0.48, only two sets of valid parallel experimental data are available,
even though we had repeated the experiment several times. Because in other parallel experiments, the SRH of the droplet is
lower than the capture range of AOT, the AOT would not be able to continue the capture when the particle size decreases to
~6 μm. Therefore, the actual SRH may be a bit lower at this pH, but this does not affect the results we discuss later. With a
decrease in pH, ammonium sulfate transforms into ammonium bisulfate. Our results are consistent with the hypothesis that
ammonium bisulfate exhibits a weaker salting out effect compared to ammonium sulfate and thus hinders the ability of organic
matter to precipitate out of the solution (Losey et al., 2018). The MRH values at pH 6.53, 5.21, 2.70, 1.19 and 0.48 were 87.6%,
89.5%, 87.3%, 83.9% and 83.5%, respectively, and are generally higher than corresponding SRH, especially in the low pH
range (<5.00). The SRH was higher than the MRH at pH 6.53, which was abnormal because a lower SRH is commonly
expected due to the activation barrier. We do not have a specific explanation for this phenomenon, while it should be noted
that the observed values were relatively close to each other, indicating that the higher SRH at pH 6.53 might potentially be
attributed to experimental error. The hysteresis between SRH and MRH existed because the SRH process has an activation
barrier while the MRH process does not, and lower RH is needed for the aerosol droplet to overcome the activation barrier to
form two phases (Freedman, 2020). Similar results were also observed in HEXT/AS and HEXD/AS systems. Additionally,
the pH-dependent SRHs obtained in this study were compared to those reported by Losey et al. (2018), as depicted in **Fig. 5a**.
It is worth mentioned that the solute concentration used in our study (50g/L) is comparable to Losey et al. (2018) (5.0 wt%),
allowing for meaningful comparison of results. Overall, the SRHs of 3-MGA obtained in this study was higher than the results
of Losey et al. (2018). When the pH was lower than 3.70, in 3-MGA system, the present study followed a similar trend as the
results of Losey et al. (2018), with the SRH decreasing as the pH decreased. However, when the pH was greater than 3.70, our
study showed an opposite trend compared to the results of Losey et al. (2018). The observed discrepancy may be attributed to
the distinct ambient conditions of the droplets. The laser levitation, resulting in a spherical morphology, while the optical
microscopy involves substrate deposition, leading to a morphology resembling a spherical crown (Tong et al.,2022; Zhou et
al., 2014). The underlying reasons for these differences are currently unclear, and further investigations are needed.

In addition to 3-MGA, we also studied two organic/AS systems to investigate how acidity affects SRH and MRH of aerosols of differing composition. These results are shown in **Fig. 5** and tabulated in **Table 2**. The separation diameter (SD) of 3-MGA/AS ranges from 7.23μm to 9.74μm, with a corresponding separation refractive index (SRI) ranging from 1.362 to 1.515. For HEXT/AS, the SD ranges from 9.01μm to 9.90μm, while the SRI ranges from 1.396 to 1.421. Lastly, the SD of HEXD/AS ranges from 7.45μm to 8.97μm, with the SRI ranging from 1.382 to 1.406. The data suggests that acidity did not have a noticeable effect on the MRH of the various systems. The pH of the HEXT/AS solution without the addition of any acid was 5.11, and SA was utilized to adjust the pH to lower levels (3.14, 2.02 and 0.92). The SRH values of HEXT/AS system (O:C=0.50) decreased as the pH decreased, with values of 78.3%, 76.6%, 76.4% and 75.7% at pH values of 5.11, 3.14, 2.02 and 0.92, respectively. The trend is similar to the 3-MGA (O:C=0.67) system, and the reason why SRH decreased may be due to the acid enhancing the miscibility of organic alcohols and inorganic substances, resulting in a greater difficulty in separating the hydrophobic phase from the water-rich phase (Tong et al., 2022). Still, we observed SRH was not strongly dependent on pH for HEXT/AS, compared to 3-MGA/AS system. This is likely due to the fact that organic alcohols have a large p$K_a$ (e.g. the p$K_a$ of HEXT is 14.3) and therefore exhibit minimal ionization in the pH range studied here (Wade and Simek, 2020). Additionally, the relative molecular interactions between alcohols and water are weaker than those of acids, leading to a weaker dependence of salting out ability of AS in the HEXT/AS system. The results of Losey et al. (2018) and Tong et al. (2022) were also depicted in **Fig. 5b**. Our results differ from those of Losey et al. (2018), who observed a significant decline in SRH as the pH decreased. The specific reason for the discrepancy remains unclear, but we speculate it may due to different condition of droplet. In contrast to the findings of Tong et al. (2022), our study observed a less pronounced trend in the values of SRH, and a narrower range in the distribution of SRH compared to literature values. The difference in OIR between this study (1:1) and Tong et al. (2022) (2:1) may account for the discrepancy in SRH. Previous studies (Ma et al., 2021; Stewart et al., 2015; Song et al., 2012) indicated that OIR differences could affect SRH, but SRH was not significantly dependent on OIR. The discrepancy in SRH may also be due to the variations in experimental conditions, such as laser power, experimental duration, etc. For HEXD/AS (O:C=0.33) system, the pH of the HEXD/AS solution without the addition of any acid was 5.01, and SA was used to adjust the pH to lower levels (3.13, 2.71, 2.03 and 1.39). SRH decreased significantly when the pH was less than 2.00, while acidity had no significant effect on SRH when pH is greater than 2.00, with values of 79.4%, 81.0%, 77.7%, 82.8% and 70.9% at pH values of 5.01, 3.13, 2.71, 2.03 and 1.39, respectively. This phenomenon was attributed to a mechanism similar to that observed in HEXT/AS. To our knowledge, this is the first investigation on the pH-dependent phase transition of HEXD/AS at the single particle level in a contact-free environment.

The pH values of misty cloud and fog droplets typically fall within the range of 2 to 7, whereas continental and marine aerosol particles display a broader spectrum of pH values (Pye et al., 2020; Tilgner et al., 2021). Our research suggests that in real atmospheric conditions, phase separation behavior of droplets may be influenced significantly by their acidity. It is challenging to measure the droplet pH of the investigated system using AOT. However, previous studies (Coddens et al., 2019; Li et al.,

2023) have shown that at high RH (90%-100%), the difference in the pH values between droplets and bulk solution is relatively
small. Therefore, we used bulk solution pH as an indicator of pH at droplet phase transition. This study focused on volatile
organics and was conducted over a relatively long period, which may have affected our results. Nevertheless, the organic
compounds used in this study have low volatility. For instance, the vapor pressure of 3-MGA is $7.41 \times 10^{-7}$ to $2.92 \times 10^{-4}$ mmHg
(DTXSID50871000, United States Environmental Protection Agency), compare to normal volatile organic components of
atmospheric aerosol, such as 2-Methyl-1-propanol with vapor pressure of 10.5 to 16.4 mmHg (DTXSID0021759, United States
Environmental Protection Agency). Volatility information of other organics are provided in the Table S5. Also, the influence
of droplet size change in our system can be neglected. For example, as shown in **Fig. 2**, the droplet size was basically same at
the beginning and the end of the experiment at the same RH 93.0% (11.85 μm at the beginning and 11.79 μm at the end).

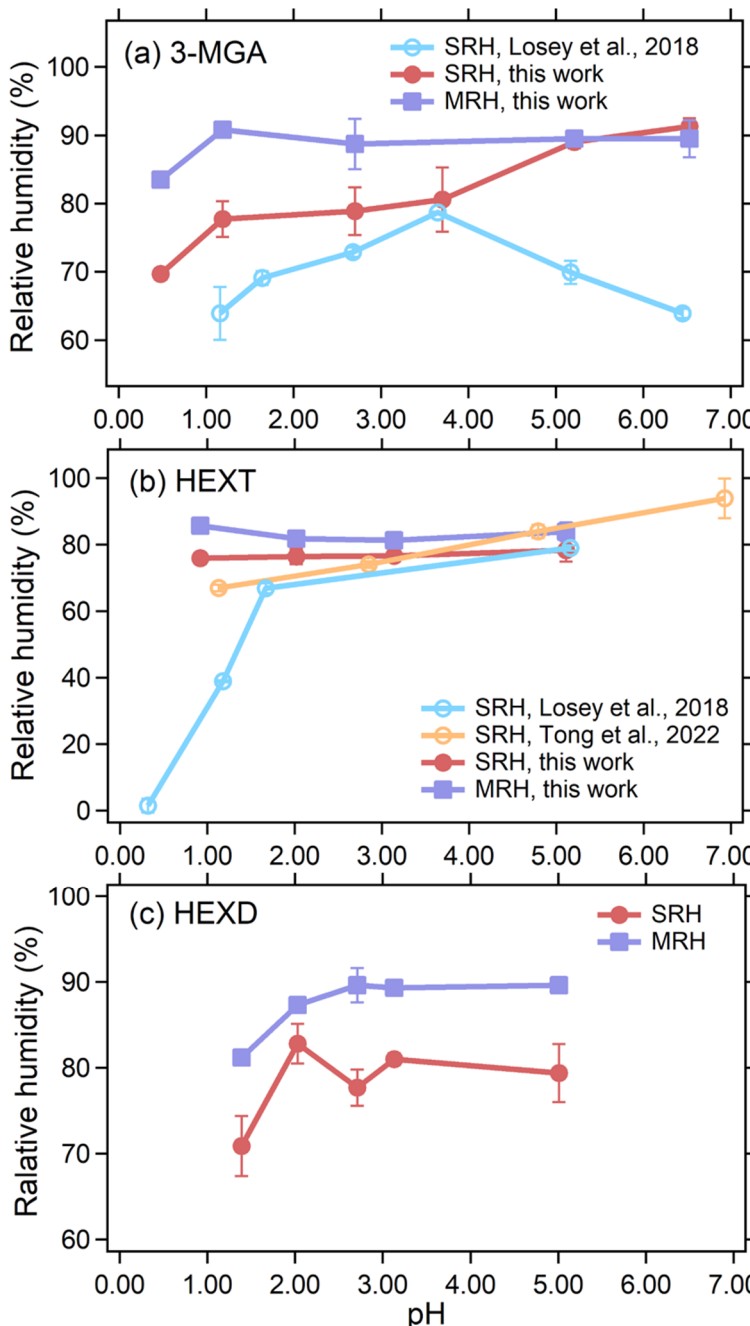


**Figure 5.** SRHs and MRHs as a function of pH for (a) 3-MGA/AS system, (b) HEXT/AS system, (c) HEXD/AS system.
Hollow circles represent data from Losey et al. (2018) and Tong et al. (2022). The error bars of SRHs and MRHs are derived
from multiple measurements.

**Table 2.** SRH information for each pH studied as well as initial diameter, separation diameter (SD), separation refractive index (SRI), MRH, mixing diameter (MD), and mixing refractive index (MRI) data.

| 3-MGA/AS system (O:C=0.67) | | | | | | | |
|---|---|---|---|---|---|---|---|
| Initial pH | Initial Diameter($\mu m$) | SRH (%) | SD ($\mu m$) | SRI ($\lambda$=650 nm) | MRH (%) | MD ($\mu m$) | MRI ($\lambda$=650 nm) |
| 0.48 | 10.97±1.57 | 69.7±0.2 | 7.23±1.72 | 1.515±0.086 | 83.5 | 6.82 | 1.540 |
| 1.19 | 11.23±1.20 | 77.7±2.6 | 8.68±2.38 | 1.454±0.100 | 90.8±0.2 | 9.08±1.64 | 1.394±0.009 |
| 2.70 | 12.02±2.94 | 78.9±3.5 | 7.88±1.21 | 1.493±0.082 | 88.7±3.7 | 6.81±2.76 | 1.506±0.094 |
| 3.70 | 10.87±1.87 | 80.6±4.7 | 7.24±1.00 | 1.491±0.088 | | | |
| 5.21 | 11.06±1.63 | 89.0±0.9 | 8.93±0.16 | 1.362±0.014 | 89.5 | 7.89 | 1.381 |
| 6.53 | 13.73±0.41 | 91.3±1.2 | 9.74±0.36 | 1.444±0.187 | 89.5±2.7 | 7.89±0.06 | 1.383±0.01 |
| HEXT/AS system (O:C=0.50) | | | | | | | |
| 0.92 | 13.52±1.6 | 75.9±0.2 | 9.90±0.76 | 1.421±0.017 | 85.7 | 10.83 | 1.420 |
| 2.02 | 12.88±1.0 | 76.4±2.3 | 9.09±0.46 | 1.409±0.007 | 81.8 | 9.34 | 1.410 |
| 3.14 | 12.31±0.8 | 76.6±1.5 | 9.01±0.47 | 1.408±0.002 | 81.3 | 9.04 | 1.409 |
| 5.11 | 13.53±0.4 | 78.3±3.4 | 9.15±0.35 | 1.396±0.014 | 83.9±2.8 | 9.04±0.73 | 1.412 |
| HEXD/AS system (O:C=0.33) | | | | | | | |
| 1.39 | 11.48±0.78 | 70.9±3.5 | 7.45±0.77 | 1.406±0.008 | 81.2 | 7.93 | 1.406 |
| 2.03 | 10.54±0.57 | 82.8±2.3 | 7.90±0.99 | 1.382±0.007 | 87.3 | 8.83 | 1.392 |
| 2.71 | 14.55±1.36 | 77.7±2.1 | 8.30±0.28 | 1.391±0.009 | 89.6±2.0 | 8.53±0.32 | 1.388±0.010 |
| 3.13 | 11.02±0.62 | 81.0±0.7 | 8.97±0.22 | 1.384±0.016 | 89.3 | 9.14 | 1.384 |
| 5.01 | 12.22±2.73 | 79.4±3.4 | 8.33±0.40 | 1.384±0.019 | 89.6±0.1 | 8.38±0.54 | 1.390±0.004 |

### 3.3 Effect of O:C on phase separation behavior in different systems

Our findings provide evidence that phase separation of droplets persists even when the organic-inorganic system is adjusted to a specific level of acidity. An important determinant of whether droplets undergo phase separation is the O:C. To illustrate this, we have included a plot in **Fig. S4**, which show cases the experimental system used in our study alongside relevant literature values. One point that needs to be declared is **Fig. S4** only plotted for systems with no additional $H_2SO_4$ or NaOH. As shown in **Fig. S4**, our findings, as well as those from previous studies (You et al., 2013; O'Brien et al., 2015), indicated that there is no correlation between the occurrence of LLPS and the hydrogen-to-carbon (H:C) ratios of the organics, which is consistent with results in previous findings (Bertram et al., 2011; Song et al., 2012). However, a clear trend was observed between LLPS occurrence and the O:C of the organic components. We observed that droplets of 3-MGA/AS, HEXT/AS and HEXD/AS systems with O:C between 0.33 and 0.67 undergo LLPS. With the decrease of water content in the droplets, two distinct phases were formed: an organic-rich phase and a salt-rich aqueous phase, under both acidic and neutral conditions. By contrast, no LLPS occurred in the GL/AS system, as shown in **Fig. S2**. In general, particles with low O:C are more prone to

undergo LLPS. This observation is consistent with the findings of Song et al. (2012) who reported that LLPS was never observed when O:C > 0.80 and always observed when O:C < 0.56.

As shown in **Fig. 2** and **Table S2**, for most spectra, WGMs remained after LLPS occurred for droplets of 3-MGA/AS. This phenomenon indicates that the droplets undergo LLPS with a core-shell morphology in most conditions, which is consistent with the prediction of Gorkowski et al. (2020). Meanwhile, morphology of phase-separated droplets containing either HEXT or HEXD were also core-shell shape mostly, as depicted in **Fig. 3/4** and **Table S3/S4**. It is attributed to the lower interfacial tension observed at higher O:C, leading to higher possibility condition for forming core-shell shaped droplets (Gorkowski et al., 2020). These findings support the idea that the O:C plays a crucial role in determining the morphology of phase-separated particles in organic/inorganic mixed aerosols.

**4 Conclusion**

The aim of this study is to investigate the effect of pH and O:C on phase transition behavior of levitated particles using the AOT. Our results show that across aerosol pH in atmospheric condition, the presence of sulfuric acid inhibited the LLPS of aerosol droplets that contained organics (3-MGA, HEXT, HEXD) and AS. Additionally, the MRHs were found to be higher than the SRHs. The O:C of phase-separating systems is 0.67, 0.50, 0.33, and by contrast, LLPS of the high O:C system (GL, O:C=1.00) did not occur. Meanwhile, the morphology of levitated aerosol particles was studied and we found that 42 out of 46 droplets that underwent LLPS for a core-shell structure. The SRH of all experimental systems ranged from ~70% to 90%. In certain cases, as the RH decreased, the droplet morphology changed from core-shell to partially engulfed, similar to the findings reported by Kucinski et al. (2020). However, as the RH further decreased, the droplet particle size became smaller than 6 μm, making it impossible to capture them using AOT. Consequently, in most instances, we were unable to observe the droplet morphology at RH levels below 70%. The results presented here provide new insights into the behavior of different types of aerosol droplets, and the findings have important implications for our understanding of physical and chemical processes that occur in the atmosphere. It is anticipated that future studies will be carried out to examine the OIR-dependent phase separation in real acidified ambient aerosols. Such research will provide insights into the morphological characteristics of real aerosols and the ways in which these characteristics influence important properties such as hygroscopicity and homogenous chemistry. Such information will be helpful in furthering our understanding of the impacts of ambient aerosols on the environment and human health.

Additionally,  in-situ pH measurement or pH estimation methods, such as the real-time AOT analysis in microdroplets reported by Boyer et al. (2020) could be combined with SRH measurements for a more accurate and comprehensive analysis.

Furthermore, our study used a surrogate for SOA instead of in situ measurements of real SOA, which can be addressed in future work using SOA generated from a smog chamber or real SOA precursors and oxidized species.

*Data availability.* The data used in this paper can be obtained from the corresponding author upon request.

*Author contributions.* YC built the instrument, performed the experiments, analyzed the data, plotted the figures, and wrote the original draft. XP conceptualized the study, contributed to instrumentation, data analysis, discussion, and reviewed the manuscript. HL and CX contributed to the instrumentation and discussion. YM contributed to the experiments and discussion. ZX, FZ contributed to the discussion and manuscript review. TCP contributed to data analysis and manuscript review. ZW administrated the project, conceptualized the study, reviewed the manuscript, and contributed to funding acquisition.

*Competing interests.* The contact author has declared that none of the authors has any competing interests.

*Financial support.* This research has been supported by the National Natural Science Foundation of China (grant nos. 91844301, 42005087, and 42005086), Key Research and Development Program of Zhejiang Province (grant nos. 2021C03165, 2022C03084), and the Fundamental Research Funds for the Central Universities (grant no. 2018QNA6008).

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
