# Peer review of "Influence of acidity on liquid-liquid phase transitions of mixed SOA"

_EGUsphere, 2023_

## Author Comment (AC1)

**Reply to comments on "Influence of acidity on liquid-liquid phase transitions of mixed SOA proxy-inorganic aerosol droplets" by Yueling Chen et al.**

**Reply to Anonymous Referee #1**

1) Chen et al. "Influence of acidity on liquid-liquid phase transitions of mixed SOA proxy-inorganic aerosol droplets" details the measurement of particle morphology and the separation relative humidity of 3 organic compounds with ammonium sulfate as a function of bulk pH.   The results differ significantly from Losey et al. 2016 and 2018 for the two overlapping systems and Tong et al. 2022 for the one overlapping system.   The explanation of the differences from previous literature is not sufficient as currently presented in the manuscript, and the work significantly overlaps with previous literature.  Specific comments are given below.   The language of the manuscript is generally clear, but should be read through with specific attention to grammar again.   This work requires substantial revision prior to possible publication.

**Response: We truly appreciate the constructive comments and suggestions raised by the referee. Those comments are valuable and very helpful for improving our paper, with important guiding significance to our studies. Below we provide a point-by-point response to individual comment. The responses are shown in brown and bold fonts, and the added/rewritten parts are presented in blue and bold fonts. Also, we have read through with specific attention to grammar again and the revised manuscript has been improved.**

**Specific Comments:**

1) **Line 33**: "Morphology can be broadly categorized into single-phase homogeneous morphology and phase separation morphology (Gorkowski et al., 2020), based on the phase state of the particle." Bertram et al. 2011 or earlier work (e.g. Ciobanu et al. 2009) may be more appropriate references for this statement.

Response: Thanks for the advice, we have revised the cited reference:

"Morphology can be broadly categorized into single-phase homogeneous morphology and phase separation morphology (Bertram et al. 2011; Ciobanu et al. 2009), based on the phase state of the particle."

2) **Line 35**: "For droplets with a phase separation morphology, the two main equilibrium morphologies are a fully engulfed (core-shell) structure and a partially engulfed structure (Freedman, 2020)." Reid et al. 2011 is a better reference for this statement.

Response: Thanks for the advice, we have revised the cited reference.

"For droplets with a phase separation morphology, the two main equilibrium morphologies are a fully engulfed (core-shell) structure and a partially engulfed structure (Reid et al. 2011)."

3) **Line 41**: "The reverse process, in which two liquid phases mix into a single homogenous liquid phase, is referred to as liquid-liquid phase mixing and the corresponding RH is the mixing RH (MRH; Gorkowski et al., 2017)." You et al. 2014 may be a better reference for this statement.

Response: Thanks for the advice, we have added the earlier reference.

**"The reverse process, in which two liquid phases mix into a single homogenous liquid phase, is referred to as liquid-liquid phase mixing and the corresponding RH is the mixing RH (MRH; You et al. 2014; Gorkowski et al., 2017)."**

4) **Line 48**: "More recently, Kucinski et al. (2021) found that submicrometer-sized aerosol particles had a lower SRH compared to micrometer-sized droplets." Both this reference and the more recent Ohno et al. should be cited (see references below).

Response: Thanks for the advice, we have added the cited references.

**"More recently, it is found that submicrometer-sized aerosol particles had a lower SRH compared to micrometer-sized droplets (Kucinski et al., 2021; Ohno et al., 2021)."**

5) **Line 57**: "Losey et al. (2018) measured the RH of phase transitions using optical microscopy and discovered that for low-pH aerosol particles (≤0.35), phase separation may be hindered by the addition of sulfuric acid." Relevant for this paper is the fact that SRH was also found to be decrease as acidity increased, and a greater change was observed for systems with lower initial SRH values. Also, both 3-methylglutaric acid and 1,2,6-hexanetriol were also studied in the cited paper. Considering you are using both sulfuric acid and sodium hydroxide to adjust the pH of your solutions, the results of Losey et al. 2016 should also be discussed and cited.

Response: Thanks for the suggestion, we have revised the manuscript.

**"Losey et al. (2018) studied six organic components and discovered that phase separation may be hindered by the addition of sulfuric acid, while the SRH of 3-methylglutaric acid/ammonium sulfate system was found to decrease with the**

addition of sodium hydroxide (Losey et al. 2016), as the deprotonation of organic component or difference in salting out ability of inorganic may change the SRH.”

6) **Line 62**: “Nevertheless, parallel experiments in this study were not conducted to accurately determine the uncertainty of the measurements.” The uncertainty of measurements of the study presented in this manuscript or in the cited manuscript?

Response: Sorry for the confusion. In Tong’s manuscript, only a single set of data was presented for each pH value. What we want to express is that parallel experiments were not conducted or clearly demonstrated to accurately determine the uncertainty of the measurements in the cited manuscript.

7) **Line 83**: The addition of sodium hydroxide changes the composition of the inorganic part of the solution, and will affect the SRH values measured.

Response: This is a very good suggestion. We used sodium hydroxide as it is a strong base, which is conducive to change the pH of the bulk solution with a minimum amount. However, it should be noted that the addition of sodium hydroxide changes the composition of the inorganic part of the solution, and may affect the SRH values measured. We have clarified in the manuscript:

“Sodium hydroxide, a strong base, allowed for pH adjustment with minimal usage (Losey et al., 2016). However, it is necessary to acknowledge that the addition of NaOH changes the composition of the inorganic part of the solution, potentially affecting the SRH values measured.”

8) **Line 152**: Insert "either" to make the sentence "When LLPS occurred, the droplets changed from a symmetrical homogeneous phase to either an asymmetrical…"

Response: Corrected.

9) **Fig. S3** is marked as GL/AS, which is not the system referred to on line 166. Then on line 177, the GL/AS system is referred to.

Response: We have revised the manuscript. Fig. S2 represents GL/AS system. Fig. S3 and Fig. S4 represent HEXT-IV and HEXD-V, respectively.

10) **Fig. 2:** Why is it clear from the WGMs that the system is phase separating into a core-shell structure rather than a partially engulfed structure? The WGMs in the phase separated region are very messy.

Response: Thanks for your comment. In the phase separation region depicted in Fig. 2c, the WGMs persist. Based on this observation, we conclude that the phase-separated system exhibits a core-shell structure rather than a partially engulfed structure. In addition, as the droplets undergo phase separation, the WGMs signal is significantly weakened, resulting in increased errors when identifying the peak positions of WGMs. As a result, the WGMs in the phase separated region are messy.

11) **Methods section:** Please clarify which systems have added $H_2SO_4$ and which systems have added NaOH. This is important for comparison to Losey et al. 2016 and 2018.

Response: We have clarified in the manuscript:

"For the 3-MGA/AS system, either SA or NaOH was utilized, while for the HEXT/AS and HEXD/AS systems, only SA was used."

**Paragraph beginning at line 186:**

12) **Comment 1:** As the pH is dropped, the pH values differ up to ~30% from Losey et al. 2016 and ~6-7% from Losey et al. 2018. The trend also differs from these two papers, with a constant decrease of SRH with decreasing pH instead of a maximum at a pH of ~pH=3.7. Certainly this can not only be due to the difference in techniques. Optical microscopy on hydrophobic substrates has been used to measure SRH in numerous papers; optical trapping has been used more rarely. The point of using a hydrophobic substrate is to minimize interactions between the particle and the substrate. SRH is generally thought to be accurate with optical microscopy, though morphology information is much less reliable. Is there a known difference in the SRH measured between these two techniques? If so, references should be provided. Also, what system was used to calibrate the SRH of the technique used in this paper? Data on calibration should be provided in the SI.

Response: Thanks for your suggestion. We admit that optical microscopy has been widely used to measure SRH in numerous papers (Ott et al., 2020; Song et al., 2012), and the results are promising when retrieving SRH. Instead, AOT is a relatively new technique for measuring the physicochemical properties of aerosol droplets, and its application in SRH measurement is limited. To our knowledge, there is only one study (Tong et al., 2022) comparing the SRH measured by these two techniques. The results showed identified discrepancies, particularly at pH values below 3. For example, at pH 4.8, the SRH values differ up to ~8%. At pH 2.8,

the discrepancy of SRH values is close. At pH 1.1, the SRH values differ up to ~40%. The authors attributed the discrepancies to the ambient conditions (e.g., levitated or deposited; spherical or spherical crown) and the Kelvin effect. Therefore, it is hard to systematically compare the measurement discrepancy.

Also, we agree that the hydrophobic substrate is able to minimize interactions between the particle and the substrate, while the morphology of the droplets may still be influenced by the contact coverslip (Zhou et al., 2014; Tong et al., 2022). Zhou et al. (2014) discussed the confocal Raman spectra analysis of mixed phthalic acid/ammonium sulfate droplets on PTFE and glass substrates, revealing the influence of salting-out effect and substrate properties on the morphology of mixed aerosols. Therefore, we speculate the different ambient conditions of the droplets may contribute to the observed discrepancy. The laser levitation, resulting in a spherical morphology, while the optical microscopy involves substrate deposition, leading to a morphology resembling a spherical crown. The underlying reasons for these differences are currently unclear, and further investigations are needed.

The applicaton of AOT in measuring SRH of aerosol droplet is limited, and there is currently no universally accepted standard substance with a recognized SRH value. Therefore, it is hard to calibrate the SRH of the AOT. We have revised the manuscript:

"The observed discrepancy may be attributed to the distinct ambient conditions of the droplets. The laser levitation, resulting in a spherical morphology, while the optical microscopy involves substrate deposition, leading to a morphology resembling a spherical crown (Tong et al., 2022; Zhou et al., 2014). The underlying

reasons for these differences are currently unclear, and further investigations are needed."

13) **Comment 2:** As the pH decreases, ammonium sulfate becomes ammonium bisulfate. The salting out ability of sulfate vs. bisulfate should be different. This is the argument made in Losey et al. 2018.

Response: Thanks for your crucial suggestions. We have revised the manuscript:

"With a decrease in pH, ammonium sulfate transforms into ammonium bisulfate. Predicted by the Hofmeister series, ammonium bisulfate exhibits a weaker salting out effect compared to ammonium sulfate and thus hinders the ability of organic matter to precipitate out of the solution (Losey et al., 2018)."

14) **Comment 3:** This manuscript reports that MRH differs from SRH for all pH values except 5.21. Losey et al. 2016 finds that MRH differs from SRH only at pH 5.17 and 6.45. MRH is the same as SRH at all other values of pH used in Losey et al. 2016 and 2018. Why is a difference observed between these two papers? Also, if MRH differs from SRH, one would expect a higher value (just as DRH>ERH because of the activation barrier required for ERH), but this is not the case for pH 6.53. What is the author's explanation of this result?

Response: Thank you for pointing this out. In principle, The MRH is higher than SRH, because the SRH process has an activation barrier while the MRH process does not, and lower RH is needed for the aerosol droplet to overcome the activation barrier to form two phases. The MRH is high in both articles across all

pH values. Therefore, we hypothesize that the difference in MRH is associated with the discrepency in SRH, which could be attributed to the distinct ambient conditions experienced by the droplets. The laser levitation, resulting in a spherical morphology, while the optical microscopy involves substrate deposition, leading to a morphology resembling a spherical crown (Tong et al., 2022), as we discussed previously.

For pH 6.53, we have conducted the parallel experiment. The SRH at this pH is higher than the MRH, and the values are relatively close to each other. We do not have a specific explaination for this phenomenon, but we suspect that it might potentially be attributed to experimental error.

15) **Comment 4:** Kucinski et al. 2019 is a study of particles < 3 microns in diameter. It is unclear based on the literature that size effects are likely for 100 micron vs. 10 micron diameter particles, as the literature (e.g. Laskina et al. 2015) compare micron to nm diameter particles.   The Kelvin effect is not an adequate explanation as it tends to affect the properties of aerosol particles starting at a size of ~100 nm and below.

Response: Many thanks for your comment. As Tong et al. (2022) explained, Kelvin effect may contributed to the discrepancy of SRH measurement. However, we agree with the reviewer, the Kelvin effect is not suitable for this case and we have consequently excluded this explanation from our discussion.

16) **Comment 5:** Both 10 micron diameter droplets and 100 micron droplets are not representative of organic aerosol found in the atmosphere, which is generally < 500 nm.

These larger droplet sizes are orders of magnitude different than the real system they are trying to simulate.    The last sentence of this paragraph should be deleted.

Response: Deleted.

**Paragraph beginning at line 209:**

17) **Comment 1:** The results for hexanetriol/ammonium sulfate differ from those of Losey et al. 2018, where a dramatic decrease in SRH with decreasing pH is observed. The trend is not similar.    The data also differ significantly from Tong et al. 2022.    At the lowest pH values, SRH was not observed in Losey et al. 2018.    What is the author's explanation of this difference between this manuscript, Losey et al. 2018, and Tong et al. 2022?    Why is only some of the data from Losey et al. 2018 plotted in Fig. 3b?

Response: Thanks for your comment. Experimental system in this work does not encompass pH values below 0.9, therefore we have only included a subset of the data from Losey et al. 2018 in previous version. We have plotted the complete result from Losey et al. 2018 in the revised Fig. 3b. In general, for HEXT/AS system, the trend between this manuscript, Losey et al. 2018, and Tong et al. 2022 is similar,which is that SRH decreases with decreasing pH. We did not find a very scientific reason to explain this difference. But we speculate that it may be due to the difference in experimental conditions. The concentration of HEXT in this work (50 g/L) is higher than concentration (2.5 wt%, about 26 g/L) of Losey et al. (2018). This difference may facilitate the precipitation of organic matter from the inorganic salts in our work. The OIR in this work is 1:1 and Tong et al. (2022) is 2:1. Previous studies (Ma et al., 2021; Stewart et al., 2015; Song et al., 2012) indicated that OIR differences could affect SRH, but SRH was not significantly dependent on OIR. The

discrepancy may also be due to the variations in experimental conditions, such as laser power, experimental duration, etc.

We have revised Fig. 3b and explaination in manuscript:

[Figure]

**Figure 5.** SRHs and MRHs as a function of pH for (a) 3-MGA/AS system, (b) HEXT/AS system, (c) HEXD/AS system. Hollow circles represent data from Losey et al. (2018) and Tong et al. (2022). The error bars of SRHs and MRHs are derived from multiple measurements.

The results of Losey et al. (2018) and Tong et al. (2022) were also depicted in Fig. 5b. "Our results differ from those of Losey et al. (2018), who observed a significant decline in SRH as the pH decreased. The specific reason for the discrepancy remains unclear, but we speculate it may due to different condition of droplet. Moreover, the concentration of HEXT in this work (50g/L) is higher than concentration (2.5 wt%, about 26 g/L) of Losey et al. (2018). The higher concentration may enhance the precipitation of organic matter from the inorganic salts in our work." In contrast to the findings of Tong et al. (2022), our study observed a less pronounced trend in the values of SRH, and a narrower range in the distribution of SRH compared to literature values. The difference in OIR between this study (1:1) and Tong et al. (2022) (2:1) may account for the discrepancy in SRH. Previous studies (Ma et al., 2021; Stewart et al., 2015; Song et al., 2012) indicated that OIR differences could affect SRH, but SRH was not significantly dependent on OIR. "The discrepancy in SRH may also be due to the variations in experimental conditions, such as laser power, experimental duration, etc."

18) **Line 232:** It is unclear how you have reached this conclusion when all of the systems in this manuscript exhibit high SRH values (> ~70% RH) at the lowest pH values studied.

Response: Based on our experimental results, we observed that high acidity hinders the phase separation of droplets. However, it is essential to acknowledge that due to the limitations of our experimental techniques, we were only able to

investigate phase separation at relative humidities above 70%. As a result, we cautiously concluded that droplets with high acidity might remain in a homogeneous phase in the real atmosphere. We admit that this statement lacks complete rigor, and we have revised the manuscript:

"The pH values of misty cloud and fog droplets typically fall within the range of 2 to 7, whereas continental and marine aerosol particles display a broader spectrum of pH values (Pye et al., 2020; Tilgner et al., 2021). Our research suggests that in real atmospheric conditions, phase separation behavior of droplets may be influenced significantly by their acidity."

19) **Line 237:** What is the volatility of each of the organic compounds used?

Response: Thanks for your comment. We have discussed the volatility of each of the organic compounds used in the manuscript:

"Nevertheless, the organic compounds used in this study have low volatility. For instance, the vapor pressure of 3-MGA is $7.41\times10^{-7}$ to $2.92\times10^{-4}$ mmHg (DTXSID50871000, United States Environmental Protection Agency), compare to normal volatile organic components of atmospheric aerosol, such as 2-Methyl-1-propanol with vapor pressure of 10.5 to 16.4 mmHg (DTXSID0021759, United States Environmental Protection Agency). Volatility information of other organics are provided in the Table S5. Also, droplet size change profile can confirm the influence of volatility of organic compounds in our system can be neglected. For example, as shown in Fig. 2, the droplet size is basically same at the beginning and the end of the experiment at the same RH 93.0% (11.85 μm at the beginning and 11.79 μm at the end)."

Table S5. Vapor pressure of organic compounds used in this study

| Compounds | Vapor pressure (mmHg) | Reference |
|---|---|---|
| GL | $1.66\times10^{-4}$ to $6.68\times10^{-3}$ | DTXSID9020663, EPA |
| 3-MGA | $7.41\times10^{-7}$ to $2.92\times10^{-4}$ | DTXSID50211649, EPA |
| | $(6.9\pm5.2)\times10^{-6}$ | Booth et al. (2010) |
| | $(5.5\pm2.0)\times10^{-6}$ | Mønster et al. (2004) |
| HEXT | $2.12\times10^{-4}$ to $1.82\times10^{-4}$ | DTXSID0041224, EPA |
| | $(1.5\pm0.15)\times10^{-6}$ | Cotterell et al. (2010) |
| | $(8.7\pm0.19)\times10^{-7}$ | Cai et al. (2015) |
| HEXD | $1.51\times10^{-2}$ to $5.27\times10^{-2}$ | DTXSID50871000, EPA |

EPA means United States Environmental Protection, https://comptox.epa.gov/ (last access: 20 April 2023).

20) **Table 2:** Should the title be "separation refractive index" and "mixing refractive index" rather than "relative index"?    These results are not discussed in the manuscript. What is the number of repeated experiments at each phase transition at each pH?    For some systems, it appears that there are only 1 or 2 repeats according to the SI.    Three or more trials should be performed.    Also, the significant figures retained are often too many considering the magnitude of the error.

**Response: Thanks for your comment. Yes, the title should be "separation refractive index" and "mixing refractive index" rather than "relative index". We have revised this information and added the discussion in the manuscript:**

**"The separation diameter (SD) of 3-MGA/AS ranges from 7.23 μm to 9.74 μm, with a corresponding separation refractive index (SRI) ranging from 1.362 to 1.515. For HEXT/AS, the SD ranges from 9.01 μm to 9.90 μm, while the SRI ranges from**

1.396 to 1.421. Lastly, the SD of HEXD/AS ranges from 7.45 μm to 8.97 μm, with the SRI ranging from 1.382 to 1.406."

   We have repeated more experiments for some systems, and have revised Fig. 3. and Table 2. For 3-MGA-I, we explain in the manuscript the reason why there are only two trials. Because in other parallel experiments, the SRH of the droplet is lower than the capture range of AOT, the AOT would not be able to continue the capture when the particle size decreases to ~6 μm. Hence, it is possible that the actual SRH at this pH is slightly lower, but this discrepancy does not impact the subsequent results we discuss. The significant figures of diameter and refractive index are determined by WGMs algorithm (Preston and Reid, 2015; Stewart et al., 2015), and the digits of precision are 3 and 4, respectively. Considering the magnitude of the error in our experiments, we have reduced the number of significant digits by one.

[Figure]

**Figure 5.** SRHs and MRHs as a function of pH for (a) 3-MGA/AS system, (b) HEXT/AS system, (c) HEXD/AS system. Hollow circles represent data from Losey et al. (2018) and Tong et al. (2022). The error bars of SRHs and MRHs are derived from multiple measurements.

**Table 2.** SRH information for each pH studied as well as initial diameter, separation diameter (SD), separation refractive index (SRI), MRH, mixing diameter (MD), and mixing refractive index (MRI) data.

| Initial pH | Initial Diameter ($\mu$m) | SRH (%) | SD ($\mu$m) | SRI ($\lambda$=650 nm) | MRH (%) | MD ($\mu$m) | MRI ($\lambda$=650 nm) |
|---|---|---|---|---|---|---|---|
| \multicolumn 3-MGA/AS system (O:C=0.67) | | | | | | | |
| 0.48 | 10.97±1.57 | 69.7±0.2 | 7.23±1.72 | 1.515±0.086 | 83.5 | 6.82 | 1.540 |
| 1.19 | 11.23±1.20 | 77.7±2.6 | 8.68±2.38 | 1.454±0.100 | 90.8±0.2 | 9.08±1.64 | 1.394±0.009 |
| 2.70 | 12.02±2.94 | 78.9±3.5 | 7.88±1.21 | 1.493±0.082 | 88.7±3.7 | 6.81±2.76 | 1.506±0.094 |
| 3.70 | 10.87±1.87 | 80.6±4.7 | 7.24±1.00 | 1.491±0.088 | | | |
| 5.21 | 11.06±1.63 | 89.0±0.9 | 8.93±0.16 | 1.362±0.014 | 89.5 | 7.89 | 1.381 |
| 6.53 | 13.73±0.41 | 91.3±1.2 | 9.74±0.36 | 1.444±0.187 | 89.5±2.7 | 7.89±0.06 | 1.383±0.01 |
| HEXT/AS system (O:C=0.50) | | | | | | | |
| 0.92 | 13.52±1.6 | 75.9±0.2 | 9.90±0.76 | 1.421±0.017 | 85.7 | 10.83 | 1.420 |
| 2.02 | 12.88±1.0 | 76.4±2.3 | 9.09±0.46 | 1.409±0.007 | 81.8 | 9.34 | 1.410 |
| 3.14 | 12.31±0.8 | 76.6±1.5 | 9.01±0.47 | 1.408±0.002 | 81.3 | 9.04 | 1.409 |
| 5.11 | 13.53±0.4 | 78.3±3.4 | 9.15±0.35 | 1.396±0.014 | 83.9±2.8 | 9.04±0.73 | 1.412 |
| HEXD/AS system (O:C=0.33) | | | | | | | |
| 1.39 | 11.48±0.78 | 70.9±3.5 | 7.45±0.77 | 1.406±0.008 | 81.2 | 7.93 | 1.406 |
| 2.03 | 10.54±0.57 | 82.8±2.3 | 7.90±0.99 | 1.382±0.007 | 87.3 | 8.83 | 1.392 |
| 2.71 | 14.55±1.36 | 77.7±2.1 | 8.30±0.28 | 1.391±0.009 | 89.6±2.0 | 8.53±0.32 | 1.388±0.010 |
| 3.13 | 11.02±0.62 | 81.0±0.7 | 8.97±0.22 | 1.384±0.016 | 89.3 | 9.14 | 1.384 |
| 5.01 | 12.22±2.73 | 79.4±3.4 | 8.33±0.40 | 1.384±0.019 | 89.6±0.1 | 8.38±0.54 | 1.390±0.004 |

21) **Fig 4:** This figure has been shown in the literature multiple times, and it is unclear what it adds to this paper. Further, these ratios at which phase separation occurs can be incorrect, depending on the system (see e.g. Ott et al. 2020). Because the main part of the paper is about aerosol pH, it is unclear why this plot is included and how it adds to the paper. Also You et al. 2013 deals with ammonium sulfate and other salts. Is only the ammonium sulfate data plotted? And for the data from this manuscript, is this only plotted for systems with no additional H2SO4 or NaOH? I recommend deleting this figure and the associated text.

Response: Thanks for your comment. We discussed Fig .4 because, iin addition to investigating systems where phase separation occurs, we also explored the phase

behavior of glycerol/AS. We found that phase separation did not occur for organics with a large O:C ratio, as examined using AOT. We agree with point that the ratios at which phase separation occurs is depending on the system. However, we only plotted for systems with no additional $H_2SO_4$ or NaOH, which means the ratios would not be affected. Yes, we only plotted the ammonium sulfate data here, because we only did the experiment with ammonium sulfate. We apologize for any confusion caused and have addressed your concern by removing Fig. 4 from the main text, relocating it to the supplement. To address your concern, we have moved Fig. 4 to the supplement while retaining the relevant discussion, as we believe it contributes to the main focus of the study. We have revised the manuscript:

"Our findings provide evidence that phase separation of droplets persists even when the organic-inorganic system is adjusted to a specific level of acidity. An important determinant of whether droplets undergo phase separation is the O:C. To illustrate this, we have included a plot in Fig. S4, which show cases the experimental system used in our study alongside relevant literature values. One point that needs to be declared is Fig. S4 only plotted for systems with no additional $H_2SO_4$ or NaOH."

22) **Line 280:** How low was SRH taken in the experiments?    Did the morphology change from core shell to partially engulfed as the RH decreased, as found in Kucinski et al. 2020 or Stewart et al. 2015?

Response: Thanks for your comment. The SRH of all experimental systems ranged from ~70% to 90%. In certain cases, as the RH decreased, the droplet morphology changed from core-shell to partially engulfed, similar to the findings

reported by Kucinski et al. in 2020. However, as the RH further decreased, the droplet particle size became smaller than 6 μm, making it is unable to persistently capture them using AOT. Consequently, in most instances, we were unable to observe the droplet morphology at RH levels below 70%. We have revised the manuscript:

[revised manuscript text omitted]

---

## Author Comment (AC2)

**Reply to comments on "Influence of acidity on liquid-liquid phase transitions of mixed SOA proxy-inorganic aerosol droplets" by Yueling Chen et al.**

**Reply to Anonymous Referee #2**

1) This work systematically studied the influence of acidity on aerosol liquid-liquid phase separation and mixing by aerosol optical tweezers coupled with Raman spectroscopy. The results showed that the higher acidity decreased the separation relative humidity (SRH), and phase separation of organic acids was more sensitive to acidity compared to alcohols. The mixing relative humidity (MRH) was found to be higher than SRH. Additionally, the results on the influence of oxygen-to-carbon ratios (O:C) showed that, while phase separation occurred in the system with O:C of 0.33, 0.50 and 0.33, no phase separation was observed in the system with high O:C (i.e., 1). These findings are interesting and important. However, I have concerns about this work as detailed in the following.

**Response: We truly appreciate the constructive comments and suggestions raised by the referee. Those comments are valuable and very helpful for improving our paper, with the important guiding significance to our studies. Below we provide a point-by-point response to individual comment. The responses are shown in brown and bold fonts, and the added/rewritten parts are presented in blue and bold fonts.**

**Major comments:**

1) **Figure1:** What are the origins of the spontaneous Raman peaks at ~3050 and 3300 cm$^{-1}$? The y axis showed the normalized intensity – please clarify to which peak the peaks were normalized to.

**Response: Thanks for the suggestion. The spontaneous Raman peaks at 3300 cm$^{-1}$ should be spurious peaks or weakened WGM peaks, because there is no spontaneous Raman peak at the same position in Fig.1b. And the origin of the spontaneous Raman peaks at ~3050 cm$^{-1}$ should be the vibration of N-H bond.The normalization of the peak is achieved by dividing it by the maximum value of the spectrum's intensity, respectively. For instance, in Fig. 1a, we observe a maximum vertical coordinate value of 4662 a.u. To normalize the light intensity of that spectrum, we divide it by 4662 and shift the entire spectrum downwards by 0.2 units. In Fig. 1b, the maximum vertical coordinate value is 871 a.u., so we divide the corresponding light intensity by 871 and shift the spectrum downwards by 0.7 units. In Fig. 1c, the maximum vertical coordinate value is 759 a.u., leading us to divide the associated light intensity by 759 and shift the spectrum downwards by 0.6 units. We have clarified it in the revised manuscript:**

"**The normalization of the peak is achieved by dividing it by the maximum value of the spectrum's intensity, respectively.**"

"**The origin of the spontaneous Raman peaks at 3300 cm$^{-1}$ and ~3050 cm$^{-1}$ are identified as the spurious or weakened WGM peaks and the vibration of N-H bond, respectively.**"

2) **Lines 141 - 142**, please explain why the area below the spontaneous Raman signal was used to normalize the Raman spectra. Normally, peak intensity is used to normalize peak intensity and peak area is used to normalize peak area.

Response: Thanks for the suggestion. During the experiment with reduced RH, we had to adjust the laser power to stablely capture droplet, which will affect the peak intensity. Therefore, we chose to normalize the area to eliminate this effect, as demonstrated by Tong et al. (2022). We have explained the reason in the revised manuscript:

"During the experiment with reduced RH, we had to adjust the laser power to ensure the stable capture of droplets, which will affect the peak intensity. To eliminate this effect, as demonstrated by Tong et al. (2022), we normalized all Raman spectra used in this study by the area below the spontaneous Raman signals."

3) **Lines 186 - 189**, can the different speciation of 3-MGA underdifferent pH conditions be the underlying reason for the different SRH. For example, under highly acidic conditions, 3-MGA mainly appears in the protonated form (conjugated acid), while under high pH conditions, the deprotonated form (conjugated base) is the major species. Different species can show different phase separation properties.

Response: Thanks for the suggestion. Indeed, the different speciation of 3-MGA under different pH conditions could be the underlying reason for the different SRH. Under low acidic conditions, 3-MGA mainly appears in the deprotonated form, which makes 3-MGA easier to precipitate. Once deprotonated, the organic component is charged and can interact with salt (i.e., ammonium sulfate) and water through ionic and ionic dipole interactions, leading to an increase in solubility. These

interactions prevent the salinization of organic components at high relative humidity (Losey et al., 2016).This trend is contrary to the phenomenon in this work that 3-MGA was more difficult to precipitate at highly acidity, so this explanation may not apply to our phenomenon. We believe that the different sorting out ability of inorganic salts under different pH is the reason for the different SRH. With a decrease in pH, ammonium sulfate transforms into ammonium bisulfate. Predicted by the Hofmeister series, ammonium bisulfate exhibits a weaker salting out effect compared to ammonium sulfate and thus hinders the ability of organic matter to precipitate out of the solution (Losey et al., 2018).

4) **Lines 234 - 236**, the statement may not be valid. Droplet pH may differ from the bulk solution pH, depending on the difference in the chemical composition between droplets and bulk solution.

Response: We agree that the droplet pH may differ from the bulk solution pH. But prevised studies (Coddens et al., 2019; Li et al., 2023) have shown that at high RH (90%~100%), the difference in the pH values between droplets and bulk solution is relatively small. Coddens et al. (2019) reported that the calculated particle pH for trapped sulfate and carbonate aerosols was generally lower than the measured bulk pH, with an average difference of about 10%.  Li et al. (2023) collected droplet Raman spectra over a relative humidity range of 100% to 90% and selected droplets with the same solute concentration as the corresponding bulk solution by using Raman proxy concentrations. The results showed that the pH of the phosphate buffered droplets and the parent bulk solution (i.e., the bulk solution with the same

solute concentration as the droplets) were identical. Therefore, we used bulk solution pH as an indicator of pH at droplet phase transition. We have revised the manuscript:

"It is challenging to measure the droplet pH of the investigated system using AOT. However, previous study (Coddens et al., 2019; Li et al., 2023) have shown that at high RH (90%~100%), the difference in the pH values between droplets and bulk solution is relatively small. Therefore, we used bulk solution pH as an indicator of pH at droplet phase transition. "

5) **Lines 237 - 239**, the statement may not be valid, as evaporation of volatile species from microdroplets have been widely observed. The authors may want to confirm that the influence in your system is neglected from the droplet size change profile. For example, a constant droplet size under a constant RH can indicate that the evaporation is neglected in your system.

Response: Thank you for the advice. We have discussed the volatility of each organics specifically in the manuscript:

"Nevertheless, the volatility of the organic compounds used in this study is low, for instance, the vapor pressure of 3-MGA is $7.41 \times 10^{-7}$ to $2.92 \times 10^{-4}$ mmHg (DTXSID50871000, United States Environmental Protection Agency), compare to normal volatile organics such as 2-Methyl-1-propanol with vapor pressure of 10.5 to 16.4 mmHg (DTXSID0021759, United States Environmental Protection Agency). Volatility information of other organics are provided in the Table S5. Also, the influence of droplet size change in our system can be neglected. For example, as shown in Fig. 2, the droplet size is basically same at the beginning and the end of the

**experiment at the same RH 93.0% (11.85 μm at the beginning and 11.79 μm at the end).”**

**SI:**

Table S5. Vapor pressure of organic compounds used in this study

| Compounds | Vapor pressure (mmHg) | Reference |
|---|---|---|
| GL | $1.66\times10^{-4}$ to $6.68\times10^{-3}$ | DTXSID9020663, EPA |
| 3-MGA | $7.41\times10^{-7}$ to $2.92\times10^{-4}$ | DTXSID50211649, EPA |
|  | $(6.9\pm5.2)\times10^{-6}$ | Booth et al. (2010) |
|  | $(5.5\pm2.0)\times10^{-6}$ | Mønster et al. (2004) |
| HEXT | $2.12\times10^{-4}$ to $1.82\times10^{-4}$ | DTXSID0041224, EPA |
|  | $(1.5\pm0.15)\times10^{-6}$ | Cotterell et al. (2010) |
|  | $(8.7\pm0.19)\times10^{-7}$ | Cai et al. (2015) |
| HEXD | $1.51\times10^{-2}$ to $5.27\times10^{-2}$ | DTXSID50871000, EPA |

EPA means United States Environmental Protection, https://comptox.epa.gov/ (last access: 20 April 2023).

**Minor comments:**

1) Please spell out the abbreviation of OIR.

**Response: Added.**

**"The pure organic and inorganic components were dissolved in ultrapure water (Millipore, resistivity of 18.2 MΩ) to create solutions with organic-to-inorganic mass ratio (OIR) of 1:1. ”**

2) Please provide the Raman spectra of droplets with different chemical composition and assign the spontaneous Raman peak in each spectrum.

**Response: Thanks, we have added the Raman spectra of droplets with different chemical composition in the supplement.**

**SI:**

[Figure]

Figure S5. Raman spectra of GL microdroplets. The WGMs are marked by black arrows. The normalization of the peak is achieved by dividing it by the maximum value of the spectrum's intensity.

[Figure]

Figure S6. Raman spectra of HEXT-II microdroplets. The WGMs are marked by black arrows. The normalization of the peak is achieved by dividing it by the maximum value of the spectrum's intensity. The origins of the spontaneous Raman peaks at 2850 and ~3050 cm$^{-1}$ are vibration of C-H and N-H bonds, respectively.

[Figure]

Figure S7. Raman spectra of HEXD-V microdroplets. The WGMs are marked by black arrows. The normalization of the peak is achieved by dividing it by the maximum value of the spectrum's intensity. The origins of the spontaneous Raman peaks at 2850 and ~3050 $cm^{-1}$ are vibration of C-H and N-H bonds, respectively.

**Reference**

Coddens, E. M., Angle, K. J., and Grassian, V. H.: Titration of aerosol pH through droplet coalescence, J. Phys. Chem. Lett., 10, 4476-4483, https://doi.org/10.1021/acs.jpclett.9b00757, 2019.Corral Arroyo, P., David, G., Alpert, P. A., Parmentier, E. A., Ammann, M., and Signorell, R.: Amplification of light within aerosol particles accelerates in-particle photochemistry, Science, 376, 293-296, https://doi.org/10.1126/science.abm7915, 2022.

Li, M., Kan, Y., Su, H., Pöschl, U., Parekh, S. H., Bonn, M., and Cheng, Y. F.: Spatial homogeneity of pH in aerosol microdroplets, Chem, in press, https://doi.org/10.1016/j.chempr.2023.02.019, 2023.

Losey, D. J., Ott, E. J. E., and Freedman, M. A.: Effects of high acidity on phase transitions of an organic aerosol, J. Phys. Chem. A, 122, 3819-3828, https://doi.org/10.1021/acs.jpca.8b00399, 2018.

---

## Author Response (AR2)

**Reply to comments on "Influence of acidity on liquid-liquid phase transitions of mixed SOA proxy-inorganic aerosol droplets" by Yueling Chen et al.**

**Reply to Anonymous Referee #1**

1) I have read the revised version of this manuscript and the authors have answered the majority of my questions. I have some further questions and suggestions based on the response to review, which I have included below.

**Response: We truly appreciate the constructive comments and suggestions raised by the referee. Those comments are valuable and very helpful for improving our paper, with important guiding significance to our studies. Below we provide a point-by-point response to individual comment. The responses are shown in brown and bold fonts, and the added/rewritten parts are presented in blue and bold fonts.**

**Specific Comments:**

1) **Reviewer 1 #11 Methods section:** Please clarify which systems have added H2SO4 and which systems have added NaOH. This is important for comparison to Losey et al. 2016 and 2018.

Response: "For the 3-MGA/AS system, either SA or NaOH was utilized, while for the HEXT/AS and HEXD/AS systems, only SA was used."

**New Question:** Is this true for all samples, i.e. did all samples of 3-MGA/AS have either sulfuric acid or sodium hydroxide, did all samples of HEXT/AS and HEXD/AS have added sulfuric acid?

Response: It is true for all system. But in each system, there is a sample where no acid or base was added. We have revised the statement in the manuscript:

"The pH of the 3-MGA/AS solution without the addition of an acid or base was 2.70. For solutions with a lower pH (1.19 and 0.48), SA was added, while NaOH was added to solutions with a higher pH (3.70, 5.21, and 6.53) to adjust their pH levels."

"The pH of the HEXT/AS solution without the addition of any acid was 5.11, and SA was utilized to adjust the pH to lower levels (3.14, 2.02 and 0.92)."

"For HEXD/AS (O:C=0.33) system, the pH of the HEXD/AS solution without the addition of any acid was 5.01, and SA was used to adjust the pH to lower levels (3.13, 2.71, 2.03 and 1.39)."

2) **Reviewer 1 #12 Paragraph beginning at line 186 Comment 1:** Is it possible to calibrate the AOT to the DRH or ERH values of known salts to give confidence in the obtained SRH and MRH values?

Response: Measuring the DRH or ERH values of known salts using AOT is challenging. AOT is unable to capture salt droplets at low humidity because, under such conditions, the droplets become too small to be effectively trapped by the AOT. Meanwhile, the shape of the droplet particle becomes irregular due to deliquescence, which results in more difficulty in partile capture. The size range of AOT to capture droplets stably is typically between 6 μm and 20 μm in diameter (Rafferty et al., 2023).

3) **Reviewer 1 #13 paragraph beginning at line 186 Comment 2:** As the pH decreases, ammonium sulfate becomes ammonium bisulfate. The salting out ability of sulfate vs. bisulfate should be different. This is the argument made in Losey et al. 2018.

Response: "With a decrease in pH, ammonium sulfate transforms into ammonium bisulfate. Predicted by the Hofmeister series, ammonium bisulfate exhibits a weaker salting out effect compared to ammonium sulfate and thus hinders the ability of organic matter to precipitate out of the solution (Losey et al., 2018)."

**New Comment:** The Hofmeister series only lists sulfate and not bisulfate, so more accurate wording would be: "With a decrease in pH, ammonium sulfate transforms into ammonium bisulfate. Our results are consistent with the hypothesis that ammonium bisulfate exhibits a weaker salting out effect compared to ammonium sulfate and thus hinders the ability of organic matter to precipitate out of the solution (Losey et al., 2018)."

Response: Thanks for the advice, we have revised the manuscript accordingly.

4) **Reviewer 1 #14 paragraph beginning at line 186 Comment 3:** This manuscript reports that MRH differs from SRH for all pH values except 5.21. Losey et al. 2016 finds that MRH differs from SRH only at pH 5.17 and 6.45. MRH is the same as SRH at all other values of pH used in Losey et al. 2016 and 2018. Why is a difference observed between these two papers? Also, if MRH differs from SRH, one would expect a higher value (just as DRH>ERH because of the activation barrier required for ERH), but this is not the case for pH 6.53. What is the author's explanation of this result?

Response: Thank you for pointing this out. In principle, The MRH is higher than SRH, because the SRH process has an activation barrier while the MRH process does not, and lower RH is needed for the aerosol droplet to overcome the activation barrier to form two phases. The MRH is high in both articles across all pH values. Therefore, we hypothesize that the difference in MRH is associated with the discrepency in SRH, which could be attributed to the distinct ambient conditions experienced by the droplets. The

laser levitation, resulting in a spherical morphology, while the optical microscopy involves substrate deposition, leading to a morphology resembling a spherical crown (Tong et al., 2022), as we discussed previously.

For pH 6.53, we have conducted the parallel experiment. The SRH at this pH is higher than the MRH, and the values are relatively close to each other. We do not have a specific explanation for this phenomenon, but we suspect that it might potentially be attributed to experimental error.

**New Comment:** It would be helpful to future readers to add to add some text to the manuscript regarding the SRH and MRH values at pH 6.53, as MRH should occur at higher RH values than SRH.

**Response: Thanks for the advice, we have added some text to the manuscript regarding the SRH and MRH values at pH 6.53:**

**"The SRH was higher than the MRH at pH 6.53, which was abnormal because a lower SRH is commonly expected due to the activation barrier. We do not have a specific explanation for this phenomenon, while it should be noted that the observed values were relatively close to each other, indicating that the higher SRH at pH 6.53 might potentially be attributed to experimental error."**

5) **Reviewer 1 #17 paragraph beginning at line 209:** I agree with most of this response with the exception of the line (this is in the response statement as well as the edits to the manuscript): "The concentration of HEXT in this work (50 g/L) is higher than concentration (2.5 wt%, about 26 g/L) of Losey et al. (2018). This difference may facilitate the precipitation of organic matter from the inorganic salts in our work." This reasoning ignores the fact that these systems both in the optical microscope and the AOT

will equilibrate to the surrounding RH, so the initial concentration of the solution is not generally the same as the concentration in the experimental droplet after equilibration.

**Response: Thanks for the suggestion, we have deleted this statement in the manuscript.**

**Reference**

Rafferty, A., Vennes, B., Bain, A., and Preston, T. C.: Optical trapping and light scattering in atmospheric aerosol science, Physical Chemistry Chemical Physics, 25, 7066-7089, 10.1039/D2CP05301B, 2023.